# Blood Immunoproteasome Activity Is Regulated by Sex, Age and in Chronic Inflammatory Diseases: A First Population-Based Study

**DOI:** 10.3390/cells10123336

**Published:** 2021-11-28

**Authors:** Ilona Elisabeth Kammerl, Claudia Flexeder, Stefan Karrasch, Barbara Thorand, Margit Heier, Annette Peters, Holger Schulz, Silke Meiners

**Affiliations:** 1Comprehensive Pneumology Center (CPC), University Hospital, Helmholtz Center Munich, Ludwig-Maximilians-University, 81377 Munich, Germany; ilona.kammerl@gmx.de (I.E.K.); scicon@t-online.de (H.S.); 2Comprehensive Pneumology Center Munich (CPC-M), German Center for Lung Research (DZL), 81377 Munich, Germany; claudia.flexeder@helmholtz-muenchen.de (C.F.); stefan.Karrasch@med.uni-muenchen.de (S.K.); peters@helmholtz-muenchen.de (A.P.); 3Institute of Epidemiology, Helmholtz Center Munich, 85764 Neuherberg, Germany; thorand@helmholtz-muenchen.de (B.T.); heier@helmholtz-muenchen.de (M.H.); 4Institute and Clinic for Occupational, Social and Environmental Medicine, University Hospital, Ludwig-Maximilians-University, 80539 Munich, Germany; 5German Center for Diabetes Research (DZD), 85764 Munich, Germany; 6KORA-Study Center, University Hospital Augsburg, 86156 Augsburg, Germany; 7Institute for Medical Information Processing, Biometry and Epidemiology, Ludwig-Maximilians-University, 81377 Munich, Germany; 8Research Center Borstel, Leibniz Lung Center, 23845 Sülfeld, Germany; 9Airway Research Center North (ARCN), German Center for Lung Research (DZL), 23845 Sülfeld, Germany; 10Institute of Experimental Medicine, Christian-Albrechts University Kiel, 24118 Kiel, Germany

**Keywords:** immunoproteasome, population-based study, peripheral blood mononuclear cells, chronic obstructive pulmonary disease (COPD), age, sex, asthma, cardiovascular disease, diabetes

## Abstract

Dysfunction of the immunoproteasome has been implicated in cardiovascular and pulmonary diseases. Its potential as a biomarker for predicting disease stages, however, has not been investigated so far and population-based analyses on the impact of sex and age are missing. We here analyzed the activity of all six catalytic sites of the proteasome in isolated peripheral blood mononuclear cells obtained from 873 study participants of the KORA FF4 study using activity-based probes. The activity of the immuno- and standard proteasome correlated clearly with elevated leukocyte counts of study participants. Unexpectedly, we observed a strong sex dimorphism for proteasome activity with significantly lower immunoproteasome activity in women. In aging, almost all catalytic activities of the proteasome were activated in aged women while maintained upon aging in men. We also noted distinct sex-related activation patterns of standard and immunoproteasome active sites in chronic inflammatory diseases such as diabetes, cardiovascular diseases, asthma, or chronic obstructive pulmonary disease as determined by multiple linear regression modeling. Our data thus provides a conceptual framework for future analysis of immunoproteasome function as a bio-marker for chronic inflammatory disease development and progression.

## 1. Introduction

The immunoproteasome is a specialized type of proteasome constitutively expressed in lymphoid cells [1,2]. In immune cells, the three immunoproteasomal catalytic subunits, i.e., low molecular mass protein (LMP) 2 or β1i, multicatalytic endopeptidase complex-like 1 (MECL1 or β2i) and LMP7 (or β5i), are replacing their constitutive counterparts, i.e., β1, β2, and β5, which are constitutively expressed in all non-immune cells of the organism. The catalytic sites of the immunoproteasome are also inducible in non-immune cells by cytokines such as interferon (IFN)-γ and tumor necrosis factor (TNF)-α [3]. Cytokine-mediated induction of the immunoproteasome in parenchymal cells is an intrinsic part of the adaptive immune response to virus infections where they serve as important producers of antigenic peptides for presentation on MHC I molecules [4,5]. In lymphoid cells, the immunoproteasome is also a crucial regulator of immune cell function. Immunoproteasome activity is required for Th1 differentiation, survival, and proliferation of T cells upon virus infection, maturation, and survival of B and plasma cells, and secretion of proinflammatory cytokines [2,6,7,8,9]. In addition, immunoproteasome function controls innate immune responses such as macrophage polarization and dendritic cell programs [10,11]. Immunoproteasome activation also emerges as part of a broader response to stress such as protein and oxidative stress [12,13,14,15]. Thus, any alteration in immunoproteasome function in lymphoid cells will have a broad impact on immune cell function. Moreover, intermediate types of proteasomes exist containing a mix of standard and immunoproteasome catalytic subunits. These may have distinct functions in health and disease [16,17,18,19,20].

Immunoproteasome function is dysregulated in multiple chronic diseases. Activation of the immunoproteasome has been reported as a pathomechanistic feature of autoimmune disorders with elevated expression of immunoproteasome subunits in circulating lymphocytes of patients with Sjögren’s Syndrome, myositis, and nephropathies [21,22,23,24]. Similarly, we recently observed that immunoproteasome activity is activated in peripheral blood mononuclear cells (PBMCs) of patients with severe chronic obstructive pulmonary diseases (COPD), a major tobacco smoke related lung disease [25]. In contrast, proteasome and immunoproteasome function were inhibited in lungs of patients with severe COPD [26,27] and by exposure to cigarette smoke in vitro and in vivo [26,28,29,30]. Immunoproteasome dysfunction is also a feature of cardiovascular diseases such as dilated cardiomyopathy, cardiac failure, and atherosclerosis and has been correlated with disease progression [31,32]. The use of immunoproteasome knockout mouse models and specific immunoproteasome inhibitors strongly suggest a causal role for the immunoproteasome for the development of cardiac hypertrophy, hypertensive cardiac remodeling, autoimmune-mediated myocarditis and atherosclerotic lesion formation [33,34,35,36,37,38]. In diabetes, impaired overall proteasome function has been observed in beta cells of the pancreas of type I diabetes patients and in experimental models [39,40,41]. The immunoproteasome, however, was induced in pancreatic islets [42,43] and might be part of a protective response to confine cytokine-mediated inflammatory signaling and autoimmune CD8^+^ T cell responses [43,44]. Of note, SNP polymorphisms in the LMP7 gene were found to be associated with a higher risk for HLA-haplotype-dependent risk for type I diabetes [44]. We here hypothesized that profiling of immunoproteasome activity in circulating blood cells might serve as a biomarker for severity and progression of these major chronic inflammatory diseases.

So far, only few explorative clinical studies have assessed the clinical value of proteasome profiling as a biomarker. One example is the analysis of blood or bone-marrow derived samples of patients with multiple myeloma to determine response to treatment with approved proteasome inhibitors, and development of resistance [45,46,47]. Other studies analyzed blood plasma or bronchioalveolar lavage for an altered amount and activity of extracellular proteasomes as a potential biomarker for acute clinical conditions such as preeclampsia, burn injury, acute ischemic stroke, and acute respiratory stress syndrome [48,49,50,51,52]. Large scale population-based studies on intracellular proteasome activity in circulating lymphocytes, however, are missing.

In the present study, we report the profiling of all catalytic activities of the standard and the immunoproteasome in peripheral mononuclear cells in the population-based Cooperative Health Research in the Region of Augsburg (KORA) study. Our pioneering study reveals an unexpected sex-related regulation of immunoproteasome activity as well as distinct activation of standard and immunoproteasome subunits in chronic inflammatory disorders such as diabetes, cardiovascular diseases, asthma, and COPD.

## 2. Materials and Methods

### 2.1. Study Population

The present study is embedded within the Cooperative Health Research in the Region of Augsburg (KORA), a series of independent population-based studies from the general population living in the region of Augsburg, Southern Germany [53,54]. Data for the current analysis was derived from the KORA FF4 study which is the 2nd follow-up study of the KORA S4 study comprising 4261 adults recruited in 1999/2001 as an independent population-based sample. The KORA FF4 study comprises 2279 study participants, examined between 2013 and 2014. Proteasome profiling was performed in the subpopulation of 924 subjects who underwent lung function testing in the KORA F4 study, the 1st follow-up study. The study was approved by the responsible ethics committee of the Bavarian Medical Association, Munich, Germany. The investigations were carried out in accordance with the Declaration of Helsinki and written informed consent was obtained from all participants.

A standardized interview was used to obtain baseline information on sociodemographic variables, self-reported physician diagnosis of common chronic diseases such as cardiovascular disorders, namely hypertension, myocardial infarction and stroke, COPD, asthma or diabetes, smoking status (never, current, ex-smoker), and medication use. Blood pressure was measured, and hypertension was defined as blood pressure greater or equal to 140/90 mmHg or use of antihypertensive medication given that participants were aware of having hypertension. Participants were classified as having asthma upon a doctor’s diagnosis. Lung function was assessed by standard spirometry, performed in line with the ATS/ERS recommendations [55]. Derived indices were forced expiratory volume in 1 s (FEV1) and forced vital capacity (FVC), defined as the maximum value within all valid maneuvers, and the ratio of both, FEV1/FVC. COPD was defined based on lung function measurements and categorized as no COPD (FEV1/FVC ≥ 70%), GOLD 1 (FEV1/FVC < 70% and FEV1 GLI percent predicted ≥ 80%), GOLD 2 (FEV1/FVC < 70% and FEV1 GLI percent predicted 50–79%), and GOLD 3 (FEV1/FVC < 70% and FEV1 GLI percent predicted 30–49%) [56]. Data on height and weight were collected in a standardized physical examination and body mass index (BMI) was calculated as the ratio of weight in kilogram and height in meter squared.

### 2.2. Blood Collection and PBMC Isolation

Valid blood samples were obtained from 924 subjects. PBMCs were isolated from 0.5 mL EDTA-blood 5–8 h after blood withdrawal using Biocoll density gradient solution (Biochrom). A total of 0.5 mL of blood was mixed with 2 mL PBS and carefully layered onto 5 mL Biocoll solution. Samples were centrifuged at 800× *g* for 30 min at room temperature. The interphase containing PBMCs was transferred to a new tube and cells were pelleted at 800× *g* for 10 min at room temperature. The cell pellet was resuspended in 1.8 mL PBS and centrifuged again with the same settings. The supernatant was discarded, and the cell pellet was stored at −80 °C.

### 2.3. Protein Extraction

Cell pellets were lysed with 40 µL native lysis buffer (50 mM Tris HCl, pH 7.5, 2 mM DTT, 5 mM MgCl_2_, 10% Glycerol, 2 mM ATP, 0.05% digitonin, cOmplete^TM^ protease inhibitor (Roche Diagnostics, Basel, Switzerland)) and incubated on ice for 15 min. Afterwards, the lysate was cleared by 15 min centrifugation at 15,000× *g* rpm at 4 °C. The supernatant was transferred to a new tube and protein concentration was determined using the Bradford method according to manufacturer’s instructions (Quick Start Bradford 1x Dye Reagent, Bio-Rad, Hercules, CA, USA) with BSA standard dilutions.

### 2.4. Activity-Based Probe Labeling

Lysates were incubated with activity-based probes MV151 (labeling all six proteolytically active β-subunits), LW124 (labeling β1 and LMP2), and MVB127 (labeling β5 and LMP7) [57]. Samples were prepared in batches of 22 samples in parallel. For every sample, two sets of protein lysates were prepared, one for MV151 and one for both LW124 and MVB127 as they differ in their fluorescence and can be analyzed in parallel. Then, 10 µg of protein lysates were diluted to 20 µL total volume with native lysis buffer without digitonin and protease inhibitors. In total, 5 µL of 5× ABP mix were added: MV151 (final concentration 0.5 µM MV151) and a combination of LW124 (0.25 µM) and MVB127 (1 µM). Samples were incubated on a shaker with 300 rpm in the dark at 37 °C. Afterwards, 5 µL of 6× Laemmli buffer (50% *v/v* glycerol, 300 mM Tris·HCl, 6% *w/v* SDS, 325 mM DTT, 0.1% *w/v* bromophenol blue, pH 6.8) were added to stop the reaction. Samples were stored at −80 °C until the gel run.

### 2.5. Gel Run

A total of 12 µL (i.e., 4 µg protein) of each sample were used per run. Four batches of 22 samples each were run in parallel at 120 V for 120 min in the dark on 12 or 18% Criterion TGX Stain-Free Precast Gel (Bio-Rad) in running buffer (25 mM Tris base, 190 mM glycin, 0.1% *w/v* SDS, pH 8.3) together with Protein Marker VI (AppliChem, Darmstadt, Germany) and three “dummy samples” to stabilize the gel run and prevent edge effects. Directly after the run, gels were scanned using the Typhoon TRIO Variable Mode Imager (GE, Munich, Germany). Images were taken at 450 PTM and 50 μm pixel resolution with fluorescence Cy3/TAMRA for ABPs MV151 and MVB127 while the Cy2 fluorescence channel was used for LW124. After the scan, the UV-inducible protein stain in the gels was activated for 5 min on the Gel Doc EZ Imager (Bio-Rad) and total protein stain was imaged.

### 2.6. Gel Analysis

Intensities of the bands were analyzed using ImageLab Software (Version 6.0.1, Bio-Rad). To normalize for protein content, the strongest band with an approx. molecular weight of 40 kDa was used as an internal reference band.

### 2.7. Data Normalization

Proteasome measurements were normalized to the protein content of the strongest band. Therefore, the original measurements were divided by the specific strongest band protein content. Furthermore, batch-specific median intensity normalization was conducted. In a first step, the batch-specific median value was calculated for each batch and proteasome parameter. In a next step, the values of the corresponding batch were divided by the batch-specific median for each parameter (i.e., normalizing the signals to the batch-specific median).

### 2.8. Statistical Analyses

Median, mean, and corresponding standard deviation (SD) or percentages and frequencies (%, *n/N*) were used to describe study characteristics. Differences between groups were tested using Wilcoxon rank sum test or Kruskal–Wallis test. In addition, pairwise Wilcoxon rank sum test was used for pairwise comparisons of more than two groups.

Linear regression models were used to analyze the association between lifestyle and sociodemographic variables as well as diseases with proteasome parameters. Due to skewed distributions, the proteasome parameters were log2 transformed for the regression analyses. After log2 transformation, all values were within the range of mean ± 5 *SD. Models with different adjustment were calculated. The basic model included sex, age, BMI, and leukocyte cell count as covariates of interest. In the extended model, diabetes, hypertension, MI/stroke, asthma, COPD, and smoking were additionally included. Interaction terms with sex were tested and the regression models were further stratified by sex. The results are presented as % change with corresponding confidence interval. The % change can be interpreted as the percent increase or decrease in the outcome variable for each one unit increase in the exposure variable. Furthermore, sensitivity analyses with adjustment for the ratio of monocyte and lymphocyte cell counts were performed. *p* values < 0.05 were considered as statistically significant. In addition, the false discovery rate (FDR) was calculated to correct for multiple testing. All analyses were performed using the statistical software package R, version 4.1.0 (R Core Team. R: A Language and Environment for Statistical Computing. Vienna, Austria. 2021. 

## 3. Results

### 3.1. Large Scale Immunoproteasome Activity Profiling in Blood Cells Using Activity-Based Probes

For large scale analysis of proteasome activities, we used a set of activity-based probes (ABPs) that have distinct binding specificities to the different active sites of the proteasome kindly provided by the lab of Hermen Overkleeft [57]. These ABPs bind to the active-site threonine of the respective catalytic subunits and label them covalently with a fluorescent dye. The number of proteolytically active proteasome subunits is quantified by fluorescent detection upon separation of labeled proteasome subunits by SDS-PAGE (Figure 1a). MV151 was used to label all standard and immuno-subunits of the proteasome and to discriminate the β2 and MECL1 activities, while LW124 and MVB127 specifically bind to the β1/LMP2 and β5/LMP7 subunits, respectively (Figure 1a). With this labeling approach, we were able to obtain information on the total activity of the proteasome, the activity of the single catalytic subunits as well as of the ratio of the respective standard versus immune subunits in a single sample (Figure 1a). We previously established the use of ABPs in our lab to detect activation of immunoproteasome activity upon inflammatory signaling and its inhibition by cigarette smoke [26,58]. For a detailed use of ABPs for proteasome activity profiling, the reader is referred to the excellent overview provided by Hewings et al. [59]. Here, we chose a combination of broad-spectrum (MV151) and subunit-specific ABPs (LW124 and MVB127) which can be applied to freshly prepared native cell extracts to differentially resolve all active sites of the proteasome. We applied this method to profile the activity of the proteasome in peripheral blood mononuclear cells (PBMCs) of more than 850 subjects.

We first established a standardized workflow for sample processing and analysis (summarized in Appendix A): (1) PBMCs were isolated according to a standard-operation procedure (SOP) over the course of one year and stored as frozen pellets at −80 °C until use. (2) Native protein extraction, protein determination and ABP labeling were performed in batches of 22 samples and lysates were stored at −80 °C until gel-based analysis using an optimized SOP. (3) Batches of labeled samples were run on two separate gels using commercially precast gels that contain an UV-activatable protein staining dye. Due to the distinct fluorophore labeling of the ABPs, this approach allowed us to resolve the β2/MECL1 and total activity on one gel and the β1/LMP2 and β5/LMP7 related activities on the second gel using two distinct fluorescence channels (Figure 1a). Upon capture of the ABP-derived fluorescent signals, gels were imaged at UV light that activated the in-gel protein stain to normalize the ABP signal from each sample to its total protein content. (4) Fluorescent ABP signals and total protein stain were densitometrically quantified. Signals were normalized to the respective protein loading using the strongest band of the protein stain, and in a second step to the median of the total activities of the gel in order to reduce batch-to-batch variability. This procedure allowed for comparative analysis of the 40 batches of PBMC samples. Each part of our workflow was accompanied by stringent quality control (for details see Appendix A). Samples that did not pass the quality control were omitted from further analysis. Appendix A shows examples for the three different ABPs and total protein staining including the densitometric analysis mask.

With this comprehensive analysis, we were able to obtain data on the relative number of each of the six active sites as well as on the ratio of the respective standard versus immunoproteasome subunits for each PBMC sample. In addition, we used the sum of all bands obtained with the MV151 ABP to quantify the total activity of the proteasome per sample as this ABP covalently labels all active subunits of the proteasome. In total, we obtained at least one activity data set from 873 study participants (Appendix A. Those also had complete information on age, sex, BMI, and leukocyte cell count.

Our optimized workflow for the analysis of more than 850 blood samples resulted in a robust detection of proteasome activity in blood mononuclear cells as exemplarily shown for a batch of 22 samples in Figure 1b. The different active sites of the proteasome as detected by the set of three ABPs were well resolved on the gel. The signals were quite homogenous indicating that our optimized workflow reduced variability due to technical handling issues. In accordance with published data [1,2], PBMCs contain mainly active immunoproteasome subunits. Activity of LMP7 was very high while β5 activity was very low. Activity of the immunosubunit MECL1 was about 3-fold higher in PBMCs than its counter standard β2 site. The activity of the β1 and LMP2 pair of catalytic subunits was detected at a ratio of about 1:1 (Figure 1b). This comparative analysis of all catalytic proteasome subunits in a large set of blood samples thus suggests the existence of intermediate types of immunoproteasomes in isolated blood mononuclear cells which almost exclusively contain LMP7, but MECL1 and LMP2 to different extents.

### 3.2. Immunoproteasome Activity Correlates with Leukocyte Cell Count

Using this data set on the proteasome activities in peripheral blood monocytes, we first analyzed whether the total leukocyte numbers or the cellular composition of the isolated PBMCs influences immunoproteasome activity. For that, we used the differential blood count data that were available for 873 study participants (Appendix A). Using a correlation analysis matrix, we observed a weak (Spearman correlation coefficient 0.04–0.32) but significant positive correlation of all standard and immunoproteasome activities as well as of the total proteasome activity with the absolute numbers of leukocytes, monocytes, and lymphocytes present in the blood (Figure 2). In this matrix, the color intensity and the size of the circles are proportional to the correlation coefficients. The total activity and the activities of MECL1 and LMP7 correlated more strongly with absolute numbers of leukocytes, lymphocytes, and monocytes compared to the other active sites. The activity of β5 was only weakly correlated with absolute immune cell numbers. Proteasome activities did not, however, correlate with the percentage of lymphocytes, monocytes and granulocytes in the blood (Appendix A) and only weakly with the absolute numbers of granulocytes (Figure 2). These data indicate that study participants with high leukocyte, monocyte or lymphocyte numbers have higher levels of immunoproteasome activity. Differences in the relative composition of monocytes and lymphocytes, which represent the main blood cell populations enriched in PBMCs, did not affect immunoproteasome activity. Our correlation analysis also provided some insight into the extent of co-regulation of the different catalytic activities of the proteasome in PBMCs: all distinct activities of the proteasome were positively correlated with each other (blue circles) but to a different extent (Figure 2). The corresponding standard and immunoproteasome activities β1/LMP2 and β2/MECL1 showed a high degree of correlation, while β5 and LMP7 were less strongly correlated. This was possibly due to the low activity signal obtained for β5 (Figure 1b).

### 3.3. Immunoproteasome Activity Is Regulated by Sex and Age

Table 1 summarizes the characteristics of the KORA FF4 study population used for immunoproteasome activity profiling. Based on this wealth of information we were able to characterize the distribution of the immunoproteasome activity in PBMCs according to basic population characteristics such as sex, age, and body mass index (BMI), but also related to smoking, lung function, and chronic diseases such as cardiovascular diseases, i.e., hypertension, myocardial infarction (MI) or stroke, diabetes, asthma and COPD.

In a first set of analyses, we determined whether immunoproteasome activity is affected by sex, age or BMI. In our descriptive analysis, we observed a strong effect of sex on immunoproteasome activities (Appendix A for total study, S2 for female and S3 for male study population). Of note, two out of three immunoproteasome activities, i.e., LMP2 and MECL1, as well as the β1 standard activity were significantly reduced in peripheral blood immune cells of women compared to men (Figure 3, Appendix A). In accordance, total proteasome activity in PBMCs was also lower in women (Figure 3, Appendix A). Regarding age-related regulation of proteasome activity, we observed activation of the MECL1 activity in study participants aged older than 60 years, which contributed to a significantly elevated ratio of MECL1 and β1 (Appendix A).

Given the sex-related differences in immunoproteasome activities, we re-analyzed our activity data separately for women and men. Of note, the proteasome was specifically activated in women aged above 60 (Figure 4, Appendix A) but not at all in men (Appendix A, Appendix A). Compared to women aged 55 or lower, elderly women displayed a uniform increase in almost all immuno- and standard activities as well as in total proteasome activity (Figure 4). We next analyzed the effect of BMI on proteasome activities. When comparing study participants according to their BMI classified as <25 kg/m^2^ (non-obese), 25–35 kg/m^2^ (mildly obese) or >35 kg/m^2^ (obese), we noted a prominent activation of standard and immunoproteasome activities in obese study participants as well as of the ratio of MECL1 and β2 (Appendix A). In our linear regression modeling, where we tested the association of the single proteasome parameters with BMI, this effect vanished suggesting that the alteration in activities depends on additional factors (Appendix A).

Importantly, the same basic linear regression modeling confirmed the robustness of the observed sex- and age-related regulation of immunoproteasome activities and demonstrated that the lower proteasome activity in women is unrelated to age, BMI, and blood composition and vice versa (Appendix A). Moreover, this regression analysis confirmed the strong correlation of almost all standard and immunoproteasome activities (except for β5) with leukocyte cell count, a well-established indicator of infections, inflammation, and/or immune system disorders.

### 3.4. Immunoproteasome Activity Is Differentially Activated in Chronic Inflammatory Diseases

In a next step, we investigated whether immunoproteasome activity in peripheral blood monocytes is altered in study participants that suffered from chronic inflammatory diseases, such as diabetes, cardiovascular diseases (hypertension, myocardial infarction, or stroke) and chronic lung diseases, i.e., COPD or asthma (see Table 1 for details). Of our 873 study participants, 56 had diagnosed diabetes (Table 1). While the non-adjusted comparison between diabetic and non-diabetic subjects suggested pronounced activation of almost all proteolytic activities of the proteasome except for LMP7 and β5 (Appendix A), most of these effects were lost upon adjustment for leukocyte count, sex, age, and BMI (Figure 5a, Appendix A). In the total population, only the β2 activity was significantly higher in diabetic patients. Separating the data according to the sex, revealed that only diabetic women displayed significantly higher β2 together with elevated β5 activity compared to non-diabetic women (Figure 5b,c). Importantly, this activation was robust and not influenced by other comorbidities as tested in our multiple linear regression modeling (Appendix A).

We also tested whether proteasome activities were altered in PBMCs of patients with cardiovascular diseases. Given the complex interaction of proteasome activities with sex and age, we focused our analysis on multiple linear regression modeling. As shown in Figure 5a–c, only the β2 activity was significantly activated specifically in men with hypertensive disease as defined by blood pressure of greater or equal to 140/90 mmHg or use of anti-hypertensive medication. In contrast, in study participants that had previously experienced myocardial infarction or stroke—32 in total—the β1 activity as well as the ratio of MECL1 and β2 were significantly higher specifically in women (Figure 5a–c, Appendix A). This effect was robust and not influenced by leukocyte counts, age, BMI, smoking, and comorbidities (Appendix A).

The availability of lung function data for our study participants allowed us to analyze how proteasome activity in peripheral blood mononuclear cells relates to lung health. Among the 80 participants that had doctor-diagnosed asthma, we only observed significant regulation of distinct proteasome activities when adjusting for leukocyte count, sex, age, BMI, smoking, and comorbidities (Figure 5, Appendix A). While the asthma patients in the total population displayed weak but significant activation of the β1 site (Figure 5a), this was absent in female asthma patients (Figure 5b). In male asthmatics, we noticed significant activation of the β5 activity and an elevated LMP7/β5 ratio (Figure 5c). Similarly, we noted distinct and sex-specific alterations in proteasomal activities in patients suffering from mild (GOLD stage 1) or more severe COPD (GOLD stages 2 + 3): the MECL1/β2 ratio was significantly elevated in the total and male study population (Figure 5a–c and Appendix A). Moreover, men with moderate to severe COPD had higher total proteasome activity in their PBMCs, while the β5 activity was specifically elevated in females with mild COPD (Figure 5a–c and Appendix A). We also confirmed that smoking has no effect on proteasome activity in PBMCs (Appendix A).

## 4. Discussion

Taken together, our first population-based analysis of proteasome activities in peripheral blood immune cells reveals a strong association of proteasome function with leukocyte cell counts—a well-known indicator of immune activation—and an unexpected sex-dependent regulation of standard and immunoproteasome activities. The influence of sex was evident not only in aging women but also in chronic inflammatory diseases of men and women, where clear sex-related activation patterns were observed for the proteasome. Of note, the observed distinct regulation of specific standard and immunoproteasome activities supports the concept of intermediate immunoproteasome complexes with different functions in health and disease.

### 4.1. Impact of Sex on Immunoproteasome Function

The lower activity of the immunoproteasome in peripheral blood mononuclear cells in women compared to men was unexpected. While several studies reported a sex-related association of specific immunoproteasome SNPs to be associated with autoimmune-related diseases such as psoriasis or multiple sclerosis [60,61], the uniform reduction of all immunoproteasome and the standard β1 activity in women has not been observed before. It is feasible that this diminished proteasome function in females contributes to the well-established differences in immune responses between men and women: the generally more pronounced innate and adaptive immune responses in females contribute to faster clearing of pathogens and greater vaccination efficiency in women while—as a drawback—leading to an increased susceptibility to inflammatory and autoimmune diseases [62,63]. Sex-related immune regulation has been partly attributed to the differential effects of sex hormones, such as estrogen and progesterone as female, and testosterone and androgens as male sex hormones. These act as steroid hormones to modulate multiple biological processes including immune regulation [64]. Population-wide analyses of sex-related gene expression signatures in PBMCs revealed distinct changes in the transcriptome that relate to sex-specific immune cell regulation [64,65].

A sexual dimorphism was also noted for the catalytic activities of the proteasome in the elderly and in several chronic inflammatory diseases. A recent study on proteasome function in mouse tissues suggested sexual dimorphism of proteasome activity with higher proteasome activities in female mouse tissues [66]. Similarly, Tiwari et al. described sex-related regulation of proteasome activity in mouse liver upon treatment with the anti-inflammatory drug ibuprofen [67]. These experimental finding supports our data and call for a systematic and mechanistic analysis of sex-specific regulation of proteasome function.

### 4.2. Age-Related Regulation of the Immunoproteasome

We and others previously reported upregulation of the immunoproteasome in aging. In mouse lungs, we observed transcriptional induction and activation of the immunoproteasome [68]. This was, however, not causally related to physiological lung aging as LMP2- or LMP7-depleted mice showed no change in aging phenotypes. The immunoproteasome has also been demonstrated to be activated in the brain, heart, liver, and muscle in experimental aging models [69,70,71] and in healthy aged humans [72,73,74]. In the present study, we observed robust and specific activation of all immunoproteasome activities and of the standard catalytic activity β1 in blood mononuclear cells of women aged 60 years and older. As the age range of our study participants was quite narrow (48–68 years), however, we cannot rule out that age-related regulation of proteasome activities also takes place in men when comparing to younger subjects. Similar to our findings in blood immune cells, Bellavista et al. reported elevated LMP7 expression in liver biopsies of aged women compared to men of similar age [72]. Evidence from experimental aging models in Drosophila and C. elegans also supports our finding on sex- and age-related regulation of proteasome function [75,76]. We speculate that this upregulation of immunoproteasome function in the elderly is an intrinsic part of the age-related changes of the immune system, i.e., immuno-aging [77].

### 4.3. Immunoproteasome Activation in Chronic Inflammatory Diseases

Our data demonstrate differential and sex-related activation of distinct standard and immunoproteasome activities in multiple chronic inflammatory diseases. Of note, we observed predominant activation of single standard activities of the proteasome namely of the β1 and β2 sites and of the ratios of respective immuno- and standard subunits. This activation was different in women and men and thus strongly supported our observed sex-related regulation of proteasome function. While, for example, the β2 activity was activated in men with hypertensive disease, proteasome activities were not significantly regulated in women. In contrast, women who had experienced myocardial infarction or stroke showed prominent activation of the β1 activity and an elevated ratio of MECL1/β2. The activities of a corresponding male population were not altered. A similar sex-specific regulation was observed for diabetes, asthma, and COPD.

One potential explanation for this distinct regulation of single activities could be their differential regulation in discrete immune cell types and/or upon activation of cell type specific immune responses. In our PBMC preparations, several immune cell types are mixed, namely monocytes and lymphocyte cell types, and a specific regulation of defined immune cell populations cannot be resolved. However, a previous comprehensive proteomic analysis on the protein composition of blood-derived immune cell types at baseline and upon activation demonstrated that most proteasome subunits are not only highly but also stably expressed in different immune cell populations (Appendix A) [78]. Activation of these immune cells induced concerted upregulation of almost all proteasome subunits in most immune cell types except for some T cell subtypes and DCs (Appendix A) [78]. Of note, the standard proteasome subunits were more strongly upregulated compared to the immunoproteasome subunits (Appendix A) suggesting a shift in catalytic proteasome complex composition. In sensitivity analyses, we accounted for the relative contribution of monocytes and lymphocytes in our isolated PBMC samples by adjustment in multiple linear regression modeling. These two leukocyte types represent the main cell populations enriched in PBMC preparations. We noted significant activation of additional proteasome activities in diabetic and COPD patients with advanced lung function impairment (GOLD 2-3) (Appendix A). These effects were again sex specific. This analysis allowed us to delineate the specific contribution of the relative composition of monocytes versus lymphocytes on distinct proteolytic activities of the proteasome in chronic inflammatory diseases. Our data require further analysis to address how the different proteasome activities are regulated in defined immune cell populations and upon specific immune cell activation.

One of the key questions emerging from our analysis is whether the alteration in immunoproteasome activities is an epiphenomenon or a causative factor that might contribute to the development or progression of chronic inflammatory diseases. Ample evidence suggests a causal role for single immunoproteasome subunits in cardiac diseases such as myocarditis, hypertension, and cardiac hypertrophy [35,36,37,38] as well as in diabetes [44] when using immunoproteasome-deficient mouse models or immunoproteasome inhibitors. Immunoproteasome function has also been investigated in experimental asthma models where LMP7 deficiency had some protective effects [79]. Its mechanistic role in chronic obstructive pulmonary diseases such as chronic bronchitis and emphysema formation, however, is unknown.

### 4.4. Intermediate Immunoproteasomes in Peripheral Immune Cells

Our findings support the concept of a differential regulation of standard and immunoproteasome subunits and the formation of intermediate immunoproteasome complexes which may have distinct functions in health and disease. In theory, a large variety of different 20S proteasome complexes (27 different combinations) can be formed by mixing of standard and immunosubunits, which might all have different functions in the cell. While cooperative assembly of the proteasome limits the diversity of some proteasome populations [80,81], the existence of different types of mixed complexes has been confirmed in human organs such as liver, kidney, and gut [82]. Forced expression of intermediate types of immunoproteasomes in immunoproteasome subunit-specific knockout mice or cells derived thereof causes differential immune defects in multiple experimental models [3,19,20,83,84,85,86]. Moreover, in vitro data with cells containing specific intermediate proteasome complexes indicates that MHC I antigen processing is altered as well as the response to oxidative stress [16,20]. We thus speculate that the here observed distinct regulation of single catalytic activities of the proteasome has an impact on immune cell function. This hypothesis needs to be further tested.

## 5. Conclusions

The present analysis provides first population-based evidence for alterations in standard and immunoproteasome activities in peripheral immune cells in health and disease. We would like to stress that even small changes in proteasome activity, as observed in this study, have a potential impact on cellular function due to the high abundance of proteasome complexes in the cell [87]. Our data raise multiple questions regarding the regulation of immunoproteasome activity by sex, the function of intermediate immunoproteasomes in immune cells, and with regard to immune responses. At the same time, this study provides an important resource to evaluate immunoproteasome activities in chronic inflammatory disorders either as a biomarker for disease stratification or as a potential therapeutic target for immunoproteasome inhibitor treatment. Figure 6 summarizes the main findings of our study which might be used as a starting point for future studies on the regulation of the immunoproteasome in health and disease.

## Figures and Tables

**Figure 1 cells-10-03336-f001:**
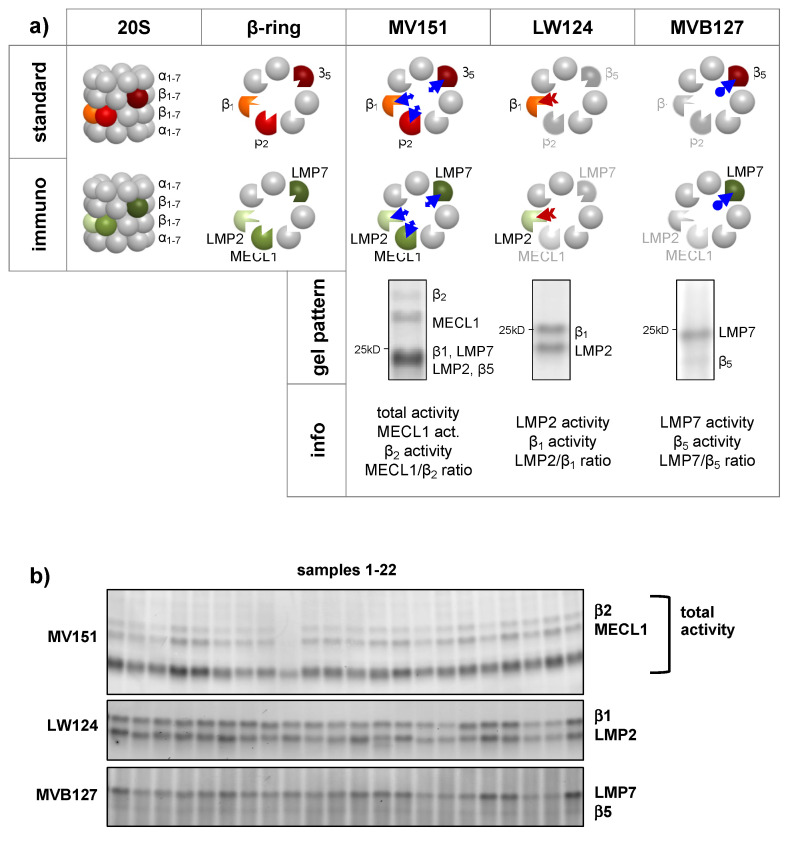
Proteasome activity profiling in peripheral blood mononuclear cells using activity-based probes (ABPs). (**a**) The 20S core particle of the proteasome is composed of four rings consisting of seven subunits each with dyad symmetry. The outer two α-rings control the opening of the gate and attachment to regulatory particles, while the inner two β-rings each comprise three different proteolytically active β-subunits: β1, β2, and β5 for the standard proteasome (present in every cell type). The immunosubunits LMP2, MECL1, and LMP7 are mainly found in lymphoid cells. With a set of three different ABPs, which bind to and fluorescently label the active centers, the six different β-subunits can be distinguished, see lower panel of the figure. MV151 labels all six proteasome subunits, and can be used for total proteasome activity assessment, but additionally MECL1 and β2 can be analyzed individually. LW124 labels LMP2 and β1, while MVB127 labels LMP7 and β5. The latter two have different fluorescent labels (here denoted in red and blue) and can be used simultaneously. For all three ABPs, the ratio of immunoproteasome subunit and their standard proteasome counterpart can also be determined. (**b**) Representative gel loaded with samples from 22 study participants and analyzed using the three activity-based probes MV151 to detect β2, MECL1 and total activities, LW124 to discriminate β1 and LMP2, and MVB127 to quantify β5 and LMP7 activities.

**Figure 2 cells-10-03336-f002:**
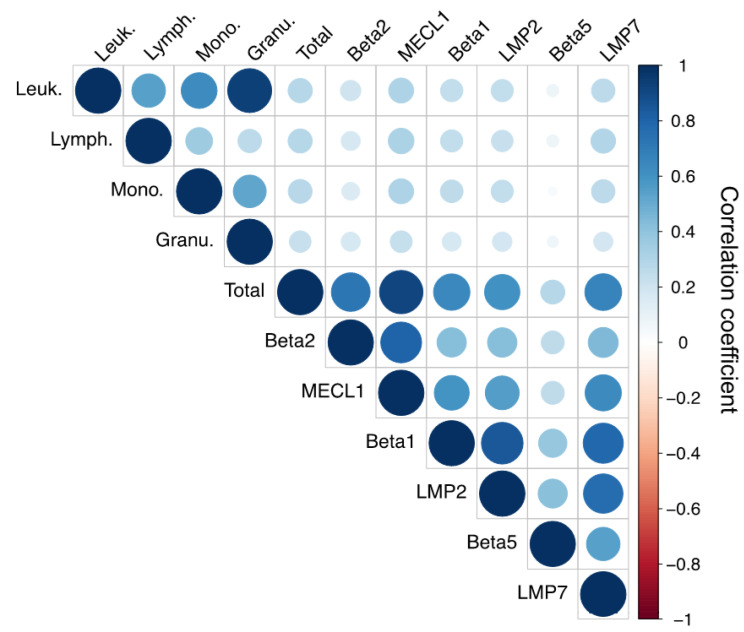
Correlation matrix of proteasome parameters and blood cell populations. Spearman’s rank correlation coefficients between standard and immunoproteasome activities with cell counts of leukocytes, lymphocytes, monocytes, and granulocytes are illustrated. Positive correlations are displayed in blue and negative correlations in red color. Color intensity and the size of the circle are proportional to the correlation coefficients.

**Figure 3 cells-10-03336-f003:**
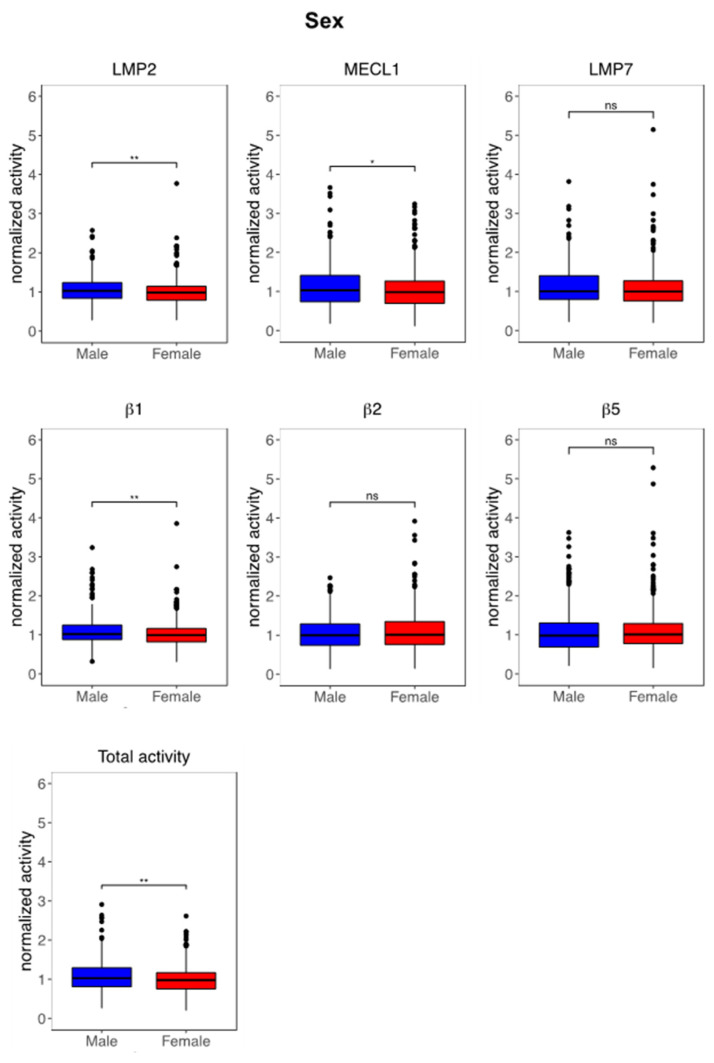
Immunoproteasome activity in PBMCs depends on sex and is lower in women. Activity-based probe (ABP) analysis of proteasome activity in peripheral blood mononuclear cells (PBMC) of male and female adults in the KORA-FF4 cohort. All samples were normalized to the batch-specific median. Normalized activities are shown as boxplots of proteasome parameters according to sex with median, 1st and 3rd quartile and whiskers indicating ± 1.5 *IQR (interquartile range). Differences between the two groups were tested using Wilcoxon rank sum test, * = *p* < 0.05 and ** = *p* < 0.01, ns = non-significant.

**Figure 4 cells-10-03336-f004:**
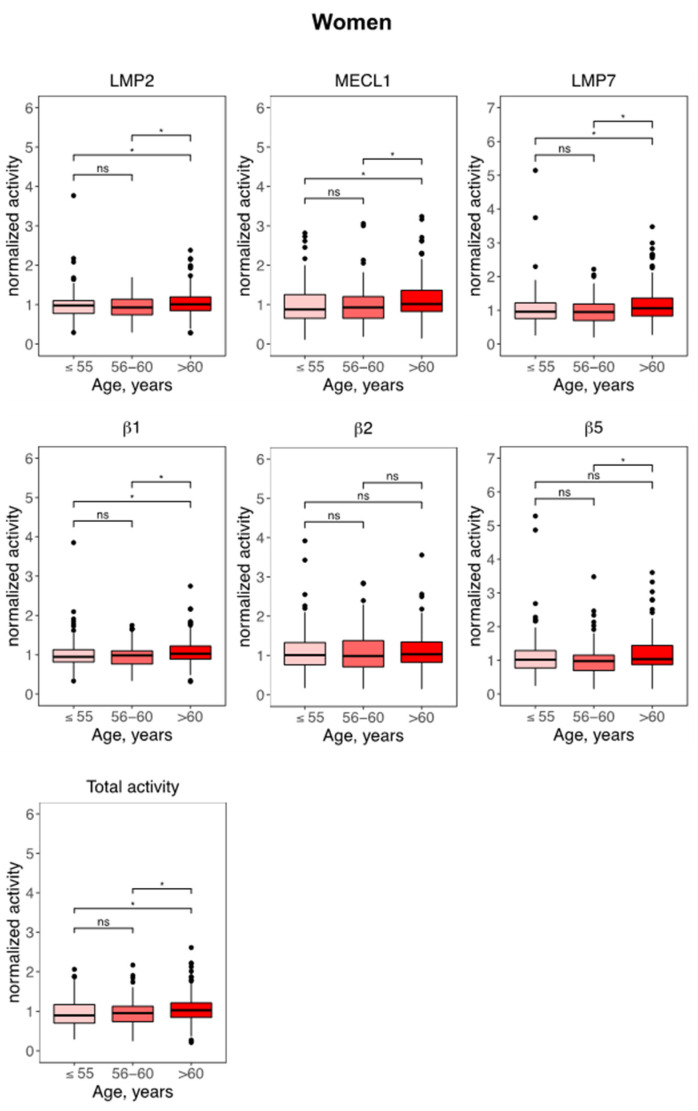
Immunoproteasome activity is distinctly activated in aged women. Activity-based probe (ABP) analysis of proteasome activity in peripheral blood mononuclear cells (PBMC) of female adults in the KORA-FF4 cohort. All samples were normalized to the batch-specific median. Age was categorized into ≤55, 56–60 and >60 years of age. Normalized proteasome activities are shown as boxplots of proteasome parameters according to age with median, 1st and 3rd quartile and whiskers indicating +/− 1.5 *IQR (interquartile range). Differences between groups were tested using pairwise Wilcoxon rank sum test. * = *p* < 0.05 and ns = non-significant.

**Figure 5 cells-10-03336-f005:**
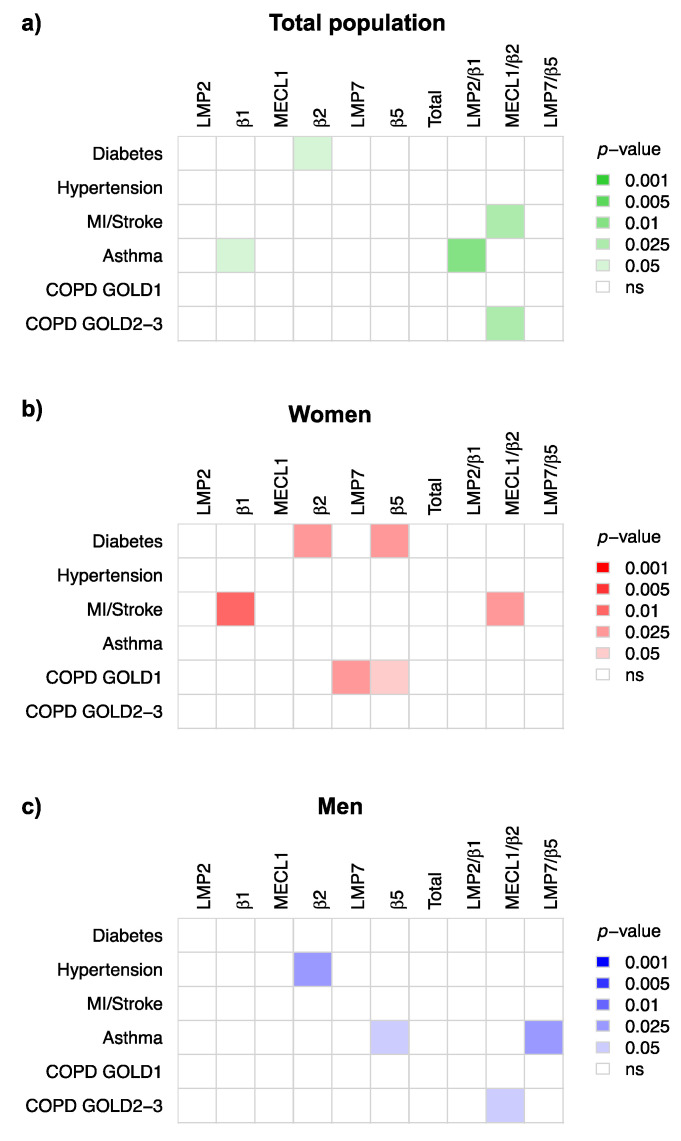
Differential regulation of proteasome activities in chronic inflammatory diseases. Multiple linear regression models for the association between diabetes, hypertension, myocardial infarction and/or stroke, asthma and COPD with log2 transformed proteasome parameters, additionally adjusted for sex, age, BMI, smoking status, and leukocyte cell count were calculated. The *p*-values resulting from the regression models based on the total study population (**a**) as well as separately for women (**b**) and men (**c**) are displayed. *p*-values above 0.05, indicating non-significant associations, are displayed in white color whereas significant associations are displayed in green, red or blue color, respectively. The color intensity increases with decreasing *p*-value.

**Figure 6 cells-10-03336-f006:**
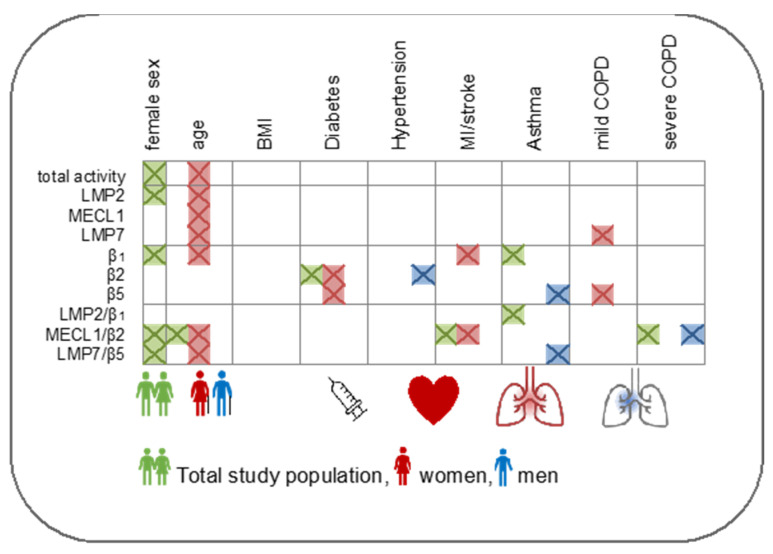
Summary of the main findings of this study with regard to the differential regulation of proteasomal activities in PBMCs depending on sex, age, BMI and chronic inflammatory diseases. Significantly regulated proteasome activities are shown in color (green: mixed population, red: female population, blue: male population) irrespective of the association strength. Non-significant associations are displayed in white color.

**Table 1 cells-10-03336-t001:** Characteristics of the KORA FF4 study population.

Parameters	Mean (SD) or %	N or *n*/*N*
Age, years	57.9 (5.7)	873
Sex		
male	47.0	410/873
female	53.0	463/873
BMI, kg/m²	27.9 (5.0)	873
FEV1, L	3.0 (0.8)	872
FVC, L	4.0 (1.0)	872
FEV1/FVC, %	75.3 (7.4)	872
COPD		
no COPD	81.0	706/872
GOLD 1	11.7	102/872
GOLD 2 or 3	7.3	64/872
CVD		
no	63.9	557/872
yes	36.1	315/872
Hypertension		
no	65.1	568/872
yes	34.9	304/872
Myocardial infarction		
no	97.7	853/873
yes	2.3	20/873
Stroke		
no	98.7	862/873
yes	1.3	11/873
Diabetes		
no	93.6	817/873
yes	6.4	56/873
Asthma DD ever		
no	90.8	793/873
yes	9.2	80/873
Smoking		
current	18.2	159/873
former	44.3	387/873
never	37.5	327/873

Abbreviations: CVD = cardiovascular diseases, i.e. hypertension, myocardial infarction, and/or stroke; DD = doctor-diagnosed; GOLD = Global Initiative for Chronic Obstructive Lung Disease; FEV1: forced expiratory volume in 1 s; FVC: forced vital capacity; GLI: Global Lung Function Initiative; COPD GOLD: no COPD = FEV1/FVC ≥ 0.7; GOLD 1 = FEV1/FVC < 0.7 and FEV1 pp (GLI) ≥ 80%; GOLD 2 = FEV1/FVC < 0.7 and FEV1 pp (GLI) 50–79%; GOLD 3 = FEV1/FVC < 0.7 and FEV1 pp (GLI) 30–49%.

## Data Availability

The data from the KORA study are subject to national data protection laws and restrictions were imposed by the Ethics Committee of the Bavarian Chamber of Physicians to ensure data privacy of the study participants. Therefore, data cannot be made freely available in a public repository. Data can be requested through an individual project agreement with KORA via the online portal KORA.PASST and requests are subject to approval by the KORA Board.

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
