# Peer review of "Blood Immunoproteasome Activity Is Regulated by Sex, Age and in Chronic Inflammatory Diseases: A First Population-Based Study"

_cells, 2021, doi:10.3390/cells10123336_

Round 1

Reviewer 1 Report

This is very well written and original study mapping the activity of proteasome across broad population with presence or absence of the most common civilization diseases.

I have only few minor points:

Line 300: The authors state that by using a correlation analysis matrix, they observed a weak but significant positive correlation with the absolute numbers of leukocytes, monocytes and lymphocytes present in the blood (Figure 2). It is not clear to me between which population was this correlation observed? Is it between total proteasome activity? Or specific subunits? Please specify.

Line 331, Table 1, the male and female are likely written in wrong columns as then the 47.0 and 53.0 numbers should be in a second column indicating percentage of respective individuals, but not the total numbers.

Author Response

This is very well written and original study mapping the activity of proteasome across broad population with presence or absence of the most common civilization diseases.

I have only few minor points:

  1. Line 300: The authors state that by using a correlation analysis matrix, they observed a weak but significant positive correlation with the absolute numbers of leukocytes, monocytes and lymphocytes present in the blood (Figure 2). It is not clear to me between which population was this correlation observed? Is it between total proteasome activity? Or specific subunits? Please specify.

R1: We thank the reviewer for pointing out that the description of the correlation matrix was not detailed enough. Accordingly, we have now provided a more detailed description of the correlations between the single immune cell populations and the total as well as single proteasomal activities. 

  1. Line 331, Table 1, the male and female are likely written in wrong columns as then the 47.0 and 53.0 numbers should be in a second column indicating percentage of respective individuals, but not the total numbers.

R2: We apologize for the mistake and have corrected the Table accordingly.

Reviewer 2 Report

The study reported in the manuscript of Prof Meiners is a real added-value in the field of aging and to derive biomarkers for common age-associated and inflammatory diseases in a sex-specific manner. The experiments are very well described and analysed, the statistical analyses are also well reported and discussed. I really support the publication of this report after very minor revision. 

1- Most of the analysis are based on the  use of specific activity-based probes provided by the team of Overkleeft , although very acknowledged in the field, authors should give more informations on these ABPs since several exists and eventually discuss the specific choice regarding the nature of biological samples since they were also used previously by the team.

2-  Finally, If possible authors should provide a concluding scheme that summarizes as complete as possible the data since a huge number of informations are given in different pathological conditions, which may help for their specific translation in term of mechanisms and/or pertinence in the immune response. This will help although partly to answer to the concluding remark lines 569-570. 

Author Response

The study reported in the manuscript of Prof Meiners is a real added-value in the field of aging and to derive biomarkers for common age-associated and inflammatory diseases in a sex-specific manner. The experiments are very well described and analysed, the statistical analyses are also well reported and discussed. I really support the publication of this report after very minor revision. 

  1. Most of the analysis are based on the  use of specific activity-based probes provided by the team of Overkleeft , although very acknowledged in the field, authors should give more informations on these ABPs since several exists and eventually discuss the specific choice regarding the nature of biological samples since they were also used previously by the team.

R3: We thank the reviewer for this comment and have accordingly included a short paragraph on the choice of the ABPs in our revised manuscript on page 5.

  1. Finally, If possible authors should provide a concluding scheme that summarizes as complete as possible the data since a huge number of informations are given in different pathological conditions, which may help for their specific translation in term of mechanisms and/or pertinence in the immune response. This will help although partly to answer to the concluding remark lines 569-570. 

R4: We would like to thank the reviewer for this excellent suggestion. Indeed, the reduction of complexity was one of the main difficulties when preparing the manuscript. We have now tried to generate a concluding scheme (Figure 6) which provides an overview on the potential use of single proteasome activities for biomarker studies.